# Recent Advances in Phage-Based Therapeutics for Multi-Drug Resistant *Acinetobacter baumannii*

**DOI:** 10.3390/bioengineering10010035

**Published:** 2022-12-27

**Authors:** Yujing Tan, Jianhui Su, Minghui Fu, Hongmei Zhang, Haiyan Zeng

**Affiliations:** School of Biomedical and Pharmaceutical Sciences, Guangdong University of Technology, Guangzhou 510006, China

**Keywords:** *Acinetobacter baumannii*, phage, endolysin, depolymerase, encapsulation, delivery

## Abstract

*Acinetobacter baumannii* is an important opportunistic pathogen common in clinical infections. Phage therapy become a hot research field worldwide again after the post-antibiotic era. This review summarizes the important progress of phage treatments for *A. baumannii* in the last five years, and focus on the new interesting advances including the combination of phage and other substances (like photosensitizer), and the phage encapsulation (by microparticle, hydrogel) in delivery. We also discuss the remaining challenges and promising directions for phage-based therapy of *A. baumannii* infection in the future, and the innovative combination of materials in this area may be one promising direction.

## 1. Introduction

*Acinetobacter baumannii* is an important opportunistic human pathogen, representing one of the most challenging hospital-acquired pathogens worldwide, causing nosocomial infections, including bacteraemia, skin and soft tissue infections, wound infections, urinary tract infections, meningitis, endocarditis, and pneumonia [1]. *A. baumannii* has a strong ability to form biofilms—a bacterial community enclosed in a matrix of self-produced extracellular polysaccharides that enables attachment to hospital medical equipment and surfaces, indeed, infections associated with the biofilms attached to surfaces are comparatively more difficult to treat [1,2]. The most common source of transmission of nosocomial *A. baumannii* infections is breathing through a ventilator, and the mortality rate of *A. baumannii*-caused ventilator-associated pneumonia (VAP) ranges from 40% to 70% [3]. As a member of the ‘ESKAPE’ (*Enterococcus faecium*, *Staphylococcus aureus*, *Klebsiella pneumoniae*, *Acinetobacter baumannii*, *Pseudomonas aeruginosa*, and *Enterobacter species*) group of pathogens, *A. baumannii* is considered a global threat to healthcare because it tends to acquire resistance to antibiotics at previously unforeseen rates [1,2]. Many multidrug-resistant (MDR) *A. baumannii* strains have developed resistance to most clinically important antibiotics, and colistin is considered a last-resort antibiotic options for treating infections caused by carbapenem-resistant gram-negative bacteria [1,2]. However, *mcr* genes which induce colistin resistance, have also been observed in *A. baumannii*, leading to a lack of treatment options in contemporary clinical practice and an undoubtedly fatal outcome [4,5]. The emergence and dissemination of MDR *A. baumannii* have thus prompted scientists to seek new antibacterial strategies and to develop new antibacterial drugs.

Phages are the most widely encountered organisms that are natural bacterial killers, and have been used in practical application since the early 20th century; the golden age of phage therapy occurred in 1920–1940s, and phages received little attention in the era of antibiotics [6]. After the post-antibiotic era, phage therapy has received renewed attention as a potential and powerful therapeutic antibacterial strategy [7,8,9]. One of the main advantages of bacteriophages over antibiotic treatments is that phages are highly specific to their targets, effectively killing the target pathogenic bacteria without affecting the commensal flora. In contrast, broad-spectrum antibiotics kill the normal bacterial flora, disrupt microbes in healthy individuals, and kill pathogens in infected individuals [2]. Phage-encoded derivatives are likewise a valuable resource for research, and both endolysins and depolymerases have been found to show superior utility than their corresponding phages [10,11,12,13].

Specificity is an advantageous feature of phage that can be exploited in the treatment of infections; however, due to the high diversity of pathogenic bacterial genomes, the limited spectrum of hosts for clinical treatment results in poor therapeutic efficacy. The phage-resistant *A. baumannii* are a very urgent issue to solve in phage therapy, and also a big obstacle for its application. In order to overcome these difficulties, the phage cocktail, phage-encoded derivatives (endolysins and depolymerases) and the combination of phage and antibiotics to control *A. baumannii* infection are developed in recent years and have been reported in many published reviews [2,14,15,16]. However, the latest advances, including the combination of phages and photosensitizer, engineered methods of applying phages and endolysins, phage encapsulation methods reported recently—have not been well summarized. In this context, the present review summarizes the latest achievements and new research progress of phage therapy in the last five years, and mainly focus on the robust application of combination therapy, engineering phages and endolysins, and advances in improving targeted delivery by using encapsulated phages in a vesicle (Figure 1). Finally, we discuss the prospects, promising directions, and potential achievements of phage therapy for *A. baumannii* in the near future.

## 2. Routine Phage-Based Therapy

### 2.1. Whole Phage and Phage Cocktail

More and more phages were found to have significant lytic activity against *A. baumannii* strain, and mainly focus on the environmental stability of the phage and its infection process and efficacy [17,18,19,20]. Single phage therapy was found to inhibit bacterial growth in a dose-dependent manner [21], and the time interval of administration also affects the phage’s therapeutic effect [22]. At the same time, the phage-resistant *A. baumannii* are also appeared very frequently [22,23]. This limitation spectrum of hosts and phage-resistant phenomenon for clinical treatment can be overcome by using phage cocktail comprising a plurality of phages varying in specific hosts and characteristics, thereby broadening the host spectrum, which is more effective than single-phage therapy [24,25,26]. Phage therapy was also reported in the treatment of a secondary MDR *A. baumannii* infection in the context of coronavirus disease 2019 (COVID-19) [27]. Four patients hospitalised with critical COVID-19 and pulmonary carbapenem-resistant *A. baumannii* infections for whom antibiotic treatment was ineffective were recruited from a COVID-19-specific intensive care unit of Shanghai Public Health Clinical Center and ultimately treated by phage therapy with a combination of phages ΦAb121 and ΦAb124 [27]. Two of the patients were discharged and survived, whereas the conditions of the other two patients improved, but they ultimately died of respiratory failure for other reasons [27].

Pater et al. prepared phage cocktails comprising three phages (ΦAb4, ΦAb7, and ΦAb14) and tested their effects in five different sets of experiments in immunocompromised rodents (6–8-week-old Swiss albino mice) with septicaemia caused by MDR *A. baumannii* Ab12 (isolated from the endotracheal tubing of a patient admitted to an ICU) to mimic real clinical situations [28]. The administration of phage cocktail after six, 12, and 24 h bacterial infection in mice resulted in the mortality ranging between 20% to 60%. However, no mortality was observed with simultaneous or prophylactic administration of phage cocktail with bacterial infection. Even after reducing the phage cocktail dose, there was still no mortality observed in these two groups [28]. This study showed that even if patients with acute infections only seek treatment at a late stage, a relatively low dose of the phage cocktail may be therapeutically beneficial.

### 2.2. Endolysins

Endolysins (lysins) are phage-encoded enzymes that degrade bacterial cell walls at the end of the phage replication cycle to release a newly assembled phage [10]. Endolysins are generally considered to be more effective against gram-positive bacteria than against gram-negative bacteria owing to the barrier of the outer membrane (OM) in the latter group. To date, some endolysins have entered different phases of clinical trials as novel antibacterial agents targeting gram-negative bacteria [29]. Endolysin is less specific than phage, one phage-encoded endolysin is capable of lysing a wide variety of bacteria. For example, the endolysin Abtn-4 from an *A. baumannii* phage D2 (vB_AbaP_D2) was reported to reduce biofilm formation and activity against several gram-positive (*Staphylococcus aureus*, *Pseudomonas aeruginosa*, *Enterococcus* spp.) and gram-negative (*A. baumannii*, *Pneumoniae klebsiella*, *Salmonella* spp.) bacteria, including phage-resistant bacterial mutant strains [30].

In addition to bacterial biofilms, the human serum also adversely affects the bactericidal efficiency of phages. A novel phage lysozyme, Abp013 of *A. baumannii* phage φAbp2, exhibited significant lytic activity against MDR strains of *A. baumannii* ATCC 17961 and tolerated the presence of up to 10% human serum to effectively kill biofilm-resident bacterial cells [31]. Along with the intrinsic ability of endolysins to penetrate the OM, co-administration with outer membrane permeabilisers (OMP), engineered lysins with biofilm-disrupting properties, and encapsulating lysins into carrier systems with OM-penetration properties are also highly effective strategies [10]. The most effective outer membrane permeabilisers reported to date include ethylenediaminetetraacetic acid (EDTA), malic acid, and citric acid [32,33,34]. The lysin LysMK34 of *A. baumannii* bacteriophage vB_AbaP_PMK34 showed good antibacterial activity against *A. baumannii* MK34 with the addition of 0.5 mM EDTA, which partially removed the dependency of the lysin on turgor pressure for osmotic lysis, enabling a high amount of LysMK34 to penetrate the OM. The use of endolysins and permeabilisers is usually limited to topical therapy. In contrast, engineered lysin and endolysin with a carrier are more suitable for application in systemic therapy (as discussed in further detail below).

Peng et al. [35] synthesised four antimicrobial peptides (AMPs) based on the endolysin LysAB2 encoded by *A. baumannii* phage ΦAB2, which exhibited potent antibacterial activity against MDR *A. baumannii.* Specifically, LysAB2 P increased the net positive charge of AMPs, reduced their hydrophobicity, and had minimal haemolytic and cytotoxic activity against normal eukaryotic cells (A549 and HaCaT cells), exerting an ideal effect in a mouse intraperitoneal infection model (infected by *A. baumannii* ATCC 17978). Thus, the results of this study showed that bacteriophage endolysins are a critical resource for the development of effective AMPs [35].

### 2.3. Depolymerases

Phage-encoded polysaccharide depolymerases can degrade bacterial surface polysaccharides to destroy biofilms, and consequent exposure of bacteria to the phage can improve their phage sensitivity, which can be applied for phage therapy and for removing *A. baumannii* biofilms from medical devices [13]. Unlike endolysins, depolymerases specifically target only a limited number of *A. baumannii* like phages, and the activities of different enzymes vary widely [36]. Thus, it is imperative to continue exploring more depolymerases that can be used to target a wide variety of bacteria and strains [11,37].

The depolymerase Dp49 of *A. baumannii* phage vB_AbaM_IME285 or human serum had no apparent bactericidal effect against *A. baumannii* Ab387 when used alone; however, when combined, a significant decrease in the bacterial count was observed in vitro. Moreover, the use of depolymerase Dp49 significantly improved the survival of mice infected with *A. baumannii* Ab387 compared with that of mice treated with phage vB_AbaM_IME285 [37]. Depolymerase Dpo48 of bacteriophage IME200 significantly improved the survival rate of *Dpo48*-pre-treated or Dpo48-treated *A. baumannii* AB1610-infected *G. mellonella*, which was 100% and 76%, respectively [11,38]. The same effect was observed in *A. baumannii*-infected mice treated with Dpo48, who showed normal serological levels and no significant histopathological changes [11]. *Acinetobacter baumannii* phage φAB6 tail spike protein (TSP) exhibits host specificity and depolymerase activity [39], and recombinantly expressed TSP significantly inhibited the biofilm formation and degraded formed biofilms [13]. The therapeutic effect of TSP in zebrafish experiments (infected by *A. baumannii* Ab-54149) was remarkable; more importantly, TSP inhibited the colonisation of *A. baumannii* on the surface of Foley catheter sections, indicating that it can be used to prevent *A. baumannii* from adhering to the surface of medical devices, a major source of hospital-acquired infections [13]. In a very recent study, several different depolymerases (DpoAB2828, DpoAB5075, DpoB8300, DpoB11911, DpoAB4932, and DpoNIPH60) encoded in the prophage regions of *A. baumannii* (AB2828, AB5075-UW, and B8300) genomes were predicted using bioinformatics approaches, which were then recombinantly produced [12]. Two of these depolymerases were specific to the capsular polysaccharides of *A. baumannii* K1 and K92 capsular types, and they significantly reduced the mortality of *G. mellonella* larvae infected with *A. baumannii* K1 and K92, respectively [12]. Thus, these enzymes show good potential as therapeutic candidates against corresponding *A. baumannii* K types.

## 3. Combination of Phage Therapy and Other Substances

### 3.1. Phages in Combination with Antibiotics

The advantages of their combination with antibiotics have been reported in many studies [40]. A synergistic effect was observed when phage vB_AbaP_AGC01 was used in combination with antibiotics (gentamicin, ciprofloxacin, and meropenem) in a *G. mellonella* infection model (infected by *A. baumannii* ATCC 16909) [41]. However, the combination of bacteriophage vB_AbaM-KARL-1 with different antibiotics showed different effects against *A. baumannii* AB01. Significantly enhanced bacterial inhibition was observed when vB_AbaM-KARL-1 was combined with colistin, no effective improvement in bacterial inhibition was observed in combination with ciprofloxacin, and complete bacterial clearance in the liquid medium occurred when combined with meropenem [42]. When the lysogenic phage (Ab177_GEIH-2000) of clinical *A. baumannii* Ab177_GEIH-2000 was mutated into a lytic phage (Ab105-2phiΔCI), the synergistic antibacterial effect with meropenem was better than that of the combination with other antibiotics in vitro, which was also verified in an experiment with *G. mellonella* in vivo [43]. Moreover, this combination reduced the incidence of phage-resistant bacteria [43].

As one of the main virulence factors of MDR *A. baumannii*, a biofilm can serve as a barrier to prevent antibiotics from entering and accessing bacterial cells, resulting in a state of antibiotic resistance. Luo et al. [44] reported that the tail fibrin of phage φAB2 and φAB6 exhibited polysaccharide depolymerase activity for the treatment of MDR *A. baumannii* A.b-M2835 and A.b-54149. Cells undergoing extracellular polymeric substance degradation altered the host susceptibility to bacterial lytic peptides (endolysin-derived peptides) and colistin. The authors further reported that the tail fibrin-modified cell wall reduces colistin attachment, leading to transient antibiotic resistance, and may increase the clinical risk in treating MDR *A. baumannii* infections [44]. In another study, colistin alone was not effective in killing *A. baumannii* MDR-AB2 in biofilms, whereas in combination with depolymerase Dpo71, more than 90% of bacterial cells in biofilms were killed [45]. In a *G. mellonella* model of *A. baumannii* MDR-AB2 infection, approximately 70% of the moths in the untreated control group died within 18 h, the mortality rate increased to 90% within 48 h, and the final survival rate in the colistin-treated group was approximately 30%; however, combination therapy with Dpo71 increased the survival rate to 80% [45]. In addition to directly inhibiting *A. baumannii*, a combination of the depolymerase Dpo71 and colistin significantly enhanced the antibiofilm activity of colistin and improved the survival rate of *A. baumannii*-infected *G. mellonella* [45].

Wang et al. [46] isolated Phab24, a phage that infects both colistin-sensitive and colistin-resistant *A. baumannii* XH198; the authors obtained phage-resistant mutant strains after incubation with its host, and they were less virulent than the parental strain in a *G. mellonella* infection model. Most importantly, increased susceptibility to colistin was observed in a phage-resistant bacterial strain that evolved in the absence of antibiotics, even though the antibiotic resistance mechanism itself remained unchanged [46]. This increase in antibiotic susceptibility was directly attributed to the phage resistance mechanism, demonstrating its potential in clinical treatment [46]. However, elucidating the complex relationship between colistin susceptibility and phage resistance warrants further attention. A recent two-stage preclinical study using a murine model of severe *A. baumannii* AB900 bacteraemia was performed to assess the in vivo bactericidal effect of the phage øFG02–ceftazidime combination [47]. The bacterial burden of the combination-treated group was significantly lower than that of the ceftazidime alone-treated groups in the first stage. However, it was not lower than that of the phage-only group until the second stage [47]. Moreover, phage øFG02 induced the in vivo evolution of *A. baumannii* towards a capsule-deficient, phage-resistant phenotype that was re-sensitised to ceftazidime [47].

### 3.2. Phages in Combination with Natural Antibacterial Agents

Natural antibacterial agents such as oils, essential oils, and other plant derivatives may serve as alternative treatments to control MDR bacterial infections, as they could be applied as additives in phage therapy medications to act in synergy with phages [48,49]. Sacha Inchi oil is extracted by pressing the seeds of *Plukenetia volubilis* Linneo, exhibiting antibacterial and anti-inflammation activities [50]. Wintachai and Voravuthikunchai [49] recently reported a synergistic effect of a novel lytic phage, vWUPSU, in combination with Sacha Inchi oil as an alternative antimicrobial agent to control MDR *A. baumannii* clinical isolate NPRCOE 160519 on planktonic cells. This combination also showed an additive effect against *A. baumannii* biofilms, indicating the potential for the development of new antibacterial and antibiofilm agents [49].

### 3.3. Phages in Combination with Photosensitizer

Photodynamic antibacterial chemotherapy (PACT) has great potential to solve serious bacterial resistance, but it suffers the lack of bacterial targeting ability [51]. Nile blue dyes (NB) are cationic photosensitizers having excellent photodynamic effect [52,53]. Phages were specific binding to their hosts [54]. Ran et al. developed a unique Nile blue photosensitizer (NB) with structural modification and ABP phage-based photodynamic antimicrobial agent (APNB) for the treatment of multi-drug resistant *A. baumannii,* and it was also used to eradicate biofilms [51]. Therefore, the constructed APNB system has dual antibacterial effects, including photodynamic therapy and phage therapy. Cytotoxicity experiments verify that APNB shows excellent biocompatibility, which is much better than that of NB. APNB provided a positive synergistic effect in the killing of pathogens in vitro and in vivo (6–8 weeks female, BALB/c mice, with 1 × 10^8^ CFU *A. baumannii* triggered skin infection) experiments, and it was first reported to effectively eradicate bacterial biofilms [51]. Moreover, the recovery from *A. baumannii* infection after APNB therapy was faster than that with antibiotics (ampicillin and polymyxin B) in vivo [51]. APNB is an interesting and powerful application direction needing pay attention to.

## 4. Engineered Phages and Endolysins

The rapid acquisition of phage resistance by bacteria could render phages already approved for therapy useless, resulting in an endless race in the search for environmental phages with high efficacy and new specificities. To avoid this challenge, many studies have confirmed the potential of engineered phages to improve the host range, enhance safety, or strengthen antibacterial activity phages [55]. In response to the problem of releasing large amounts of toxins when phages lyse their host cells, non-replication or lysis-deficient mutant phages were designed [56]. Although research on engineered phages that are effective against MDR *A. baumannii* is still limited, the theoretical concepts and application results in other bacteria indicate great potential in this field [57].

The CRISPR-Cas system has been widely used in the genetic editing of eukaryotic organisms, and there are also some successful applications in bacteria [58]. In a teaching experiment, students applied CRISPR-Cas9 to design the phage AB1 with modification of the tail protein gene sequence, which altered the binding ability of the phage to the host *A. baumannii* ABA46, resulting in lysing of the bacterial cells, and the gene-edited phage also showed promising results in the mouse test in vivo [59].

Protein engineering has been extensively used to modify modular gram-positive bacteria lysins to improve their activity, stability, and solubility [10]. The application of phage lysin PlyAB1 is limited by its thermal stability and lytic activity [60]. Based on molecular dynamics simulations and Hotspot wizard 3.0 analysis, three double-point variants, G100Q/K69R, G100R/K69R, and G100K/K69R, with significantly improved thermal stability and improved lytic activity were obtained [60]. At 45 °C, the lytic activity and half-life of the optimal variant G100Q/K69R were 1.51- and 24-fold higher than those of the wild PlyAB1, respectively. These results help us understand the structure and function of phage lysin and contribute to its application in antibiotic substitution [60]. In a recent study, Abdelkader et al. [60] leveraged the intrinsic antibacterial activity of *A. baumannii* endolysin LysMK34 by fusing the peptide cecropin A (with outer membrane-permeabilising properties) to its N-terminus via a linker of three Ala-Gly repeats, resulting in engineered LysMK34 (eLysMK34), which has a C-terminal amphipathic helix structure that enables penetrating the otherwise impermeable outer membrane barrier [60]. Compared to that of the parental lysin, eLysMK34 showed better antibacterial activity in terms of the minimum inhibitory concentration (0.45 to 1.2 μM), killing rate, and extent while reducing the dependency of the endolysin on intracellular osmotic pressure for its bactericidal efficiency. Moreover, colistin-resistant *A. baumannii* strains became susceptible to eLysMK34, and enhanced antibacterial activity was observed in complement-deactivated human serum [60]. In another study, cecropin A (CecA) was also fused to the N-terminus of putative endolysin ST01 with minor lytic activity from novel *Salmonella typhimurium* phage PBST08 [61]. The resulting CecA::ST01 has been shown to have increased bactericidal activity against many gram-negative pathogens, and the most affected target was *A. baumannii* [61]. These studies thus revealed that fusing an outer membrane-permeabilising peptide to an endolysin with intrinsic antibacterial activity results in a superior lysin with more robust antibacterial activity [61].

## 5. Delivery and Encapsulated Phage Preparation

Along with selecting and/or designing an appropriate phage or derivative for *A. baumannii* infection treatment, it is equally important to determine the most suitable route of administration, including the lung, nasal cavity, oral cavity, intravenous injection, and intraperitoneal cavity. Each of these routes is associated with different challenges in passing environmental barriers for delivery to the infection site and avoiding their degradation and clearance by the host defence system upon the entry of phage into human bodies, which will affect the therapeutic efficiency of the phage [61,62]. Encapsulation of phages and endolysins in delivery vehicles can provide targeted therapy with controlled compound delivery, surpassing chemical, physical, and immunological barriers that can inactivate and eliminate the therapeutic agents [61]. To date, phages have been encapsulated in more delivery vehicles than endolysins, including biopolymeric structures, particles, nanoparticles, a freeze-dried formulation/spray with stabilisers, hydrogels, and liposomes [61,63].

For the treatment of lung/respiratory tract infections, devices such as nebulisers, pressurised metered-dose inhalers, and powder inhalers are typically used to ensure that the drug can effectively reach the site of infection [64]. Nebulisation was proposed as an additional bacteriophage delivery modality, potentially effective for the eradication of MDR bacteria in the lungs [64,65]. In a clinical case of a 52-year-old critically ill patient with MDR *A. baumannii* respiratory infection, combination therapy with antibiotics (sulfamethoxazole/trimethoprim and tigecycline) and bacteriophages (AbW4932 and AbW4878) was used, as the choice of antibiotics was limited [66]. Initially, a combination of intravenous bacteriophage therapy with broad-spectrum antibiotics was administered for 14 days. Although the patient showed clinical improvement, the treatment failed to eradicate the MDR *A. baumannii* infection, as revealed by positive sputum cultures. Subsequently, a reactive bacteriophage diluted in normal saline delivered in a vibrating mesh nebuliser was used for nebulisation, which was combined with intravenous bacteriophage therapy along with antibiotics for another 21 days. This treatment was discontinued after two consecutive negative sputum samples were obtained, and the patient was weaned off mechanical ventilation after significant clinical improvement. This case indicated that nebulisation might be a more efficacious delivery method for directly targeting a phage to pathogens in the airways and lung parenchyma.

Encapsulation of phages aims at promoting the controlled release and enhancing viable phage persistence at the wound site. Chitosan-based hydrogels have also been used as carriers for other molecules in enhancing diabetic wound healing [67]. In a recent study, Ilomuanya et al. [68] developed encapsulated *A. baumannii* phage cocktails (ΦAB140 and ΦAB150) using chitosan and resuspended the microparticles in hydrogels to treat a chronic wound caused by MDR *A. baumannii* strains (isolated from wound samples). Diabetic rats infected with *A. baumannii* were used to evaluate the bacterial clearance efficiency of wounds, and the in vivo results showed significant wound size reduction on day 4. Moreover, in the group receiving the microparticle-encapsulated *A. baumannii* phage hydrogel, no *A. baumannii* infection was detected on days 7, 10, or 14 post-treatments, whereas infections persisted in the control groups. The formulation, including chitosan-based encapsulated phage microparticles suspended in a hydrogel, was proven to be safe, non-toxic, and effective by causing the complete clearance of infection from the wound bed. In another study, a sodium alginate-based biohydrogel was prepared, combining choline oleate as a transdermal delivery enhancer and a lytic phage cocktail (PhL_UNISO_AB-ATC_ph0035 and PhL_UNISO_AB-ATC_ph0041) to investigate its treatment potential for *A. baumannii* ATCC 19606 infection [69]. The phage preparation encapsulated by the biohydrogel exhibited potent lytic activity against *A. baumannii*, and exhibited excellent permeability in pig ear skin, which is similar to human skin [69]. Endolysin can also be encapsulated in a delivery vesicle; however, this has not yet been evaluated with respect to *A. baumannii* treatment [61].

## 6. Conclusions and Prospect

Phages are natural killers of bacteria exhibiting great potential in addressing the growing threat of MDR strains, phage cocktails have been identified as the optimal solution in clinical application, while, more cocktail combinations need to be explored to identify the cocktail with the best effect and to further elucidate the combination mechanism [70,71,72,73]. Except for phage therapy, endolysins and depolymerases can also participate in the synthesis of antibacterial peptides or other antibacterial drugs, and be applied for removing *A. baumannii* biofilms from medical devices, respectively [10,13]. However, the high specificity of depolymerases like phage limited its application.

Owing to the popularisation and application of metagenomic sequencing technology, numerous novel phages and large phage genomes have been discovered, which has given new vitality to the phage research field [12,54]. Further analysis of unknown functional proteins of *A. baumannii* phages, and the promotion of the development of phage-derived enzyme preparations are expected to improve the application effect and scope of phages. The rapid occurrence of phage resistance by bacteria have rendered the application of phage therapy essential. Therefore, basic research to uncover the detailed interaction mechanism of *A. baumannii* phages is necessary, including the characteristics of phage host receptors and the molecular mechanism of *A. baumannii* against phages, which may also provide vital theoretical support for the genetic modification and artificial synthesis of phages in clinical application.

The combination of phages and other antimicrobial substances also provides promise in *A. baumannii* therapy, which requires further experimental research to establish the best combination scheme and use strategy [47,50,70]. Phage in combination with photosensitizer is an interesting direction, and it may have great potential in fighting against multidrug-resistant *A. baumannii* and biofilm ablation in practice [51]. The safety of a single phage preparation depends not only on the safety of the phage itself but also on its preparation method [16]. Accordingly, standardising the phage preparation method is crucial, as purification of phage from bacterial hosts may result in unintended delivery of bacterial toxins such as endotoxins and/or exotoxins [71]. Several studies have expounded the importance of standardising the preparation of phage preparations, actively promoting associated research progress for application in treatment [72,73]. As the phage encapsulation method is beneficial for delivery in clinical applications and for enhancing the efficiency of phage therapy [61], more attention should be given to this approach in the therapy of *A. baumannii* infections. The innovative application of novel materials in phage-combination therapy or phage encapsulation may be a promising direction in the future.

As an ancient and effective biological control method, phage therapy has become a hot research field worldwide owing to the continual emergence of MDR bacteria, posing a setback in the development of new antibiotics.

## Figures and Tables

**Figure 1 bioengineering-10-00035-f001:**
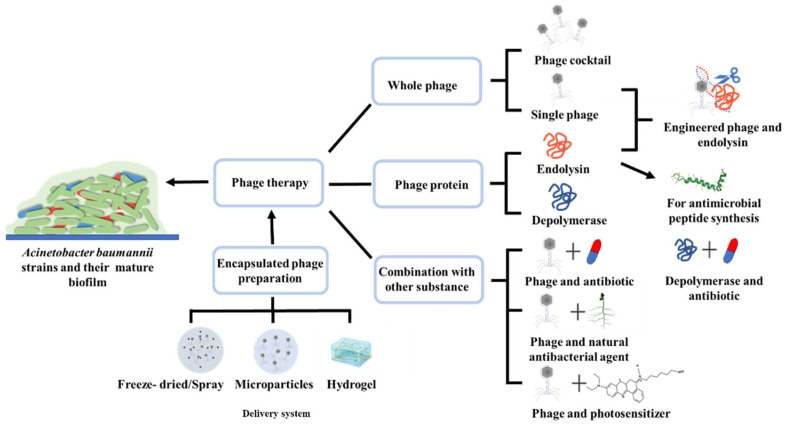
Strategies for phage therapy to control *Acinetobacter baumannii* infection.

## Data Availability

Not applicable.

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
