# Peer review of "Recent Advances in Phage-Based Therapeutics for Multi-Drug Resistant *Acinetobacter baumannii"

_bioengineering, 2022, doi:10.3390/bioengineering10010035_

Round 1

Reviewer 1 Report

In this detailed review, the authors summarized the latest achievements and new research progress of phage therapy in the last five years, and mainly focus on the robust application of combination therapy, engineering phages and endolysins, and advances in improving targeted delivery by using encapsulated phages in a vesicle. They also discussed the prospects, promising directions, and potential achievements of phage therapy for A. baumannii in the near future. The manuscript is well written and up to date, and summarizes the results of recent findings concerning the phage-based therapeutics for multidrug resistant A. baumannii. It should be of interest for readers of phage therapy. I have only some minor comments that are listed below:

1. Lines 89-92, “The mortality rate for the groups inoculated with the phage cocktail at 6, 12, and 24 h after infection was 20–60%; however, there was no mortality in the group that received the phage cocktail concomitant with bacterial stimulation or before vaccination, even after reducing the cocktail dose”. This description is confusing.

2. Line 96, change “4” to “Four”.

3. Line 362, Patents?

Author Response

In this detailed review, the authors summarized the latest achievements and new research progress of phage therapy in the last five years, and mainly focus on the robust application of combination therapy, engineering phages and endolysins, and advances in improving targeted delivery by using encapsulated phages in a vesicle. They also discussed the prospects, promising directions, and potential achievements of phage therapy for A. baumannii in the near future. The manuscript is well written and up to date, and summarizes the results of recent findings concerning the phage-based therapeutics for multidrug resistant A. baumannii. It should be of interest for readers of phage therapy. I have only some minor comments that are listed below:

  1. Lines 89-92, “The mortality rate for the groups inoculated with the phage cocktail at 6, 12, and 24 h after infection was 20–60%; however, there was no mortality in the group that received the phage cocktail concomitant with bacterial stimulation or before vaccination, even after reducing the cocktail dose”. This description is confusing.

Response: Thanks for your good suggestion. We have rewritten these sentences in line 93~97 as follows: “The administration of phage cocktail after six, 12, and 24 h bacterial infection in mice resulted in the mortality ranging between 20% to 60%. However, no mortality was observed with simultaneous or prophylactic administration of phage cocktail with bacterial infection. Even after reducing the phage cocktail dose, there was still no mortality observed in these two groups.”

  1. Line 96, change “4” to “Four”.

Response: We are sorry for this error, “4” is the number added by mistake, we have deleted it.

  1. Line 362, Patents?

Response: This section is not mandatory but may be added if there are patents resulting from the work reported in this manuscript. For there were no patents resulting from this work, so we have deleted this section. Thanks for your carefully check.

Reviewer 2 Report

(1) Check the writing carefully throughout to avoid minor mistakes.

(2) More discussion about engineering phages and endolysin should be included in section 4. 

Author Response

(1) Check the writing carefully throughout to avoid minor mistakes.

Response: Thanks for your suggestion. We have checked the writing carefully throughout and correct all minor mistakes.

(2) More discussion about engineering phages and endolysin should be included in section 4. 

Response: Thanks for your good suggestion. There are still limited papers related with engineering phages and endolysins in A. baumannii. We have added all related research in this area when we drafted our manuscript. Happily, two new papers have been published recently [Yuan Y, Li Q, Zhang S, Gu J, Huang G, Qi Q, et al. Enhancing thermal stability and lytic activity of phage lysin PlyAB1 from Acinetobacter baumannii. Biotechnol Bioeng. 2022 Oct;119(10):2731-42.; Lim J, Hong J, Jung Y, Ha J, Kim H, Myung H, et al. Bactericidal Effect of Cecropin A Fused Endolysin on Drug-Resistant Gram-Negative Pathogens. J Microbiol Biotechnol. 2022 Jun 28;32(6):816-23.].

We have added the description of the first paper in section 4 line 264~267 as follows: “The application of phage lysin PlyAB1 is limited by its thermal stability and lytic activity. Based on molecular dynamics simulations and Hotspot wizard 3.0 analysis, three double-point variants, G100Q/K69R, G100R/K69R, and G100K/K69R, with significantly improved thermal stability and improved lytic activity were obtained. At 45°C, the lytic activity and half-life of the optimal variant G100Q/K69R were 1.51- and 24-fold higher than those of the wild PlyAB1, respectively. These results help us understand the structure and function of phage lysin and contribute to its application in antibiotic substitution”;

We have added the description of the second paper in section 4 line 282~286 as follows: “In another study, cecropin A (CecA) was also fused to the N-terminus of endolysin ST01 with minor lytic activity from novel Salmonella typhimurium phage PBST08. The resulting CecA::ST01 has been shown to have increased bactericidal activity against many gram-negative pathogens, and the most affected target was A. baumannii. ”

Reviewer 3 Report

The statement in lines 44-45 is misleading. Phage-resistant bacteria appear very frequently. For this reason, phage-resistant bacteria are a very urgent issue in phage therapy. I checked the cited reference 10, which is already incorrectly reported within this reference, and no citation is given. In this reference, the authors report the appearance of resistant bacteria during treatment of Acinetobacter baumannii infections (citation [50]). I would like you to survey the literature more on phage-resistant bacteria and report in detail in this review.

Author Response

The statement in lines 44-45 is misleading. Phage-resistant bacteria appear very frequently. For this reason, phage-resistant bacteria are a very urgent issue in phage therapy. I checked the cited reference 10, which is already incorrectly reported within this reference, and no citation is given. In this reference, the authors report the appearance of resistant bacteria during treatment of Acinetobacter baumannii infections (citation [50]). I would like you to survey the literature more on phage-resistant bacteria and report in detail in this review.

Response: Thanks for your good suggestion. This sentence “One of the main advantages of bacteriophages over antibiotic treatments is that bacteria appear to develop resistance to bacteriophages at a minimal rate.” is the opinion cited from the reference 10, which was described in the first paragraph of the bacteriophage therapy section in page 38 of this reference. Unfortunately, this sentence has no exact source of quotation in reference 10. For the sake of preciseness, we have deleted this sentence in our revised manuscript.

As stated by the reviewer, the phage-resistant A. baumannii are a very urgent issue to solve in phage therapy, and also a big obstacle for its application. In order to overcome these difficulties, the phage cocktail, phage-encoded derivatives (endolysins and depolymerases) and the combination of phage and antibiotics to control A. baumannii infection are developed in recent years and have been reported in many published reviews. (We have added these sentences in line 54~58 of revised manuscript). However, the latest advances, including the combination of phages and photosensitizer, engineered methods of ap-plying phages and endolysins, phage encapsulation methods reported recently—have not been well summarized. In our manuscript, we mainly focus on these new achievements in the phage therapy. Although we did not describe the phage-resistant bacteria as separate paragraph, it had existed in the descriptions of other sections, like phages in combination with antibiotics etc. In addition, we have also added the phage-resistant A. baumannii descriptions and related references in line 76~78.

Round 2

Reviewer 2 Report

Previous comments have been addressed. I would suggest accepting the manuscript in its current version.